# Post lockdown COVID-19 seroprevalence and circulation at the time of delivery, France

Jérémie Mattern[1], Christelle Vauloup-Fellous[2], Hoda Zakaria[1], Alexandra Benachi[1], Julie Carrara[1], Alexandra Letourneau[1], Nadège Bourgeois-Nicolaos[3], Daniele De Luca[4], Florence Doucet-Populaire[3], Alexandre J. Vivanti[1] *

1 Division of Obstetrics and Gynecology, Antoine Béclère Hospital, Paris Saclay University, AP-HP, Clamart, France, 2 Division of Virology, Paul Brousse Hospital, Paris Saclay University, AP-HP, INSERM U1193, Villejuif, France, 3 Division of Microbiology, Antoine Béclère Hospital, Paris Saclay University, AP-HP, Clamart, France, 4 Division of Pediatrics and Neonatal Critical Care, Antoine Bécleère Hospital, Paris Saclay University, AP-HP, Clamart, France

* alexandre.vivanti@aphp.fr

## Abstract

### Background

To fight the COVID-19 pandemic, lockdown has been decreed in many countries worldwide. The impact of pregnancy as a severity risk factor is still debated, but strict lockdown measures have been recommended for pregnant women.

### Objectives

To evaluate the impact of the COVID-19 pandemic and lockdown on the seroprevalence and circulation of SARS-CoV-2 in a maternity ward in an area that has been significantly affected by the virus.

### Study design

Prospective study at the Antoine Béclère Hospital maternity ward (Paris area, France) from May 4 (one week before the end of lockdown) to May 31, 2020 (three weeks after the end of lockdown). All patients admitted to the delivery room during this period were offered a SARS-CoV-2 serology test as well concomitant SARS-CoV-2 RT-PCR on one nasopharyngeal sample.

### Results

A total of 249 women were included. Seroprevalence of SARS-CoV-2 was 8%. The RT-PCR positive rate was 0.5%. 47.4% of the SARS-CoV-2-IgG-positive pregnant women never experienced any symptoms. A history of symptoms during the epidemic, such as fever (15.8%), myalgia (36.8%) and anosmia (31.6%), was suggestive of previous infection.

### Conclusions

Three weeks after the end of French lockdown, SARS-CoV-2 infections were scarce in our region. A very high proportion of SARS-CoV-2-IgG-negative pregnant women, which is

**Data Availability Statement:** All relevant data are within Manuscript and Supporting Information files.

**Funding:** The authors received no specific funding for this work.

**Competing interests:** The authors have declared that no competing interests exist.

comparable to that of the general population, must be taken into consideration in the event of a resurgence of the pandemic. The traces of a past active circulation of the virus in this fragile population during the spring wave should encourage public health authorities to take specific measures for this independent at-risk group, in order to reduce viral circulation in pregnant patients.

## Introduction

The coronavirus disease 2019 (COVID-19) pandemic hit France at the end of February 2020. To date (August 29, 2020), more than 24,000,000 cases and 830,000 deaths have occurred worldwide [1]. France, and more particularly the Paris area, was one of the regions of the world most affected by the pandemic in spring 2020 [2, 3]. In order to slow down progression of the pandemic, lockdown has been decreed in many countries worldwide. French lockdown started on March 17, 2020, and ended on May 11, 2020.

Pregnancy as a severity risk factor of severe acute respiratory syndrome coronavirus 2 (SARS-CoV-2) infection has been debated, even though the third trimester seems to be a consensual risk factor [4–9]. Recent evidence in the literature suggests that infection among pregnant women with COVID-19 increases the risk of iatrogenic preterm birth, cesarean delivery and maternal complications in the postpartum period, making obstetric management of potentially infected patients a complex issue [10, 11]. In addition, it has recently been shown that the virus can be transmitted transplacentally and be responsible for neurological symptoms in newborns [12]. All of these maternal and neonatal data make it important to carefully consider the possibility of infection during pregnancy.

Lockdown has been shown to result in a very significant decrease in the incidence of COVID-19 infections in pregnant women in France [13]. The main goal of this study was to evaluate the impact of the first wave of the COVID-19 pandemic in our region, the quantity of pregnant patients exposed to the virus and the effect of lockdown on seroprevalence in our maternity ward, which has been significantly affected by the virus (more than 40 confirmed infections among pregnant women between March 12 and April 20) [8].

## Materials and methods

We conducted a prospective study at the Antoine Béclère Hospital maternity ward (Paris area, France) from May 4 (one week before the end of lockdown) to May 31, 2020 (three weeks after the end of lockdown). All patients admitted to the delivery room during this period were offered a SARS-CoV-2 serology test as well as SARS-CoV-2 RT-PCR on a nasopharyngeal sample. A questionnaire was distributed to all included women, who were asked about symptoms (fever, cough, dyspnea, myalgia, anosmia, diarrhea and rash) affecting them and/or close relatives (household residents) from January 1 to their delivery, and if there were symptoms they were asked for the date of onset. Patients were also asked whether they strictly adhered to the rules of lockdown and social distancing.

### Serology testing

Serum samples were run on the Abbott Architect instrument using the Abbott SARS-CoV-2 IgG assay (Abbott, Sligo, Ireland) following the manufacturer's instructions. The assay is a chemiluminescent microparticle immunoassay for detection in human serum or plasma of

IgG against the SARS-CoV-2 nucleocapsid protein. IgM detection was not performed. Based on our local evaluation of 500 serum samples (data not published), the result was considered negative if the IgG index value was < 0.8, equivocal if the IgG index was between 0.8 and 1.39, and positive if the IgG index value was ≥ 1.40 (0.40 is the Abbott determined positivity cut-off).

### Molecular testing

Nasopharyngeal swabs were collected from all the patients. Samples were tested by using the GeneFinder™ COVID-19 Plus Real*Amp* Kit (Elitech, Puteaux, France) on the ELITe InGenius® Platform, or the Xpert Xpress SARS-CoV-2 assay (Cepheid™, Sunnyvale, CA, USA), or the GenMark ePlex SARS-CoV-2 assay (Elitech, Puteaux, France), depending on the time of day. We adapted our algorithm of the assay choice to our workflow, to our reagent supply possibilities and to the urgency of the results for clinical management of some parturients.

The GeneFinder assay is a molecular in vitro test utilizing one-step reverse transcription real-time PCR to detect the RNA-dependent RNA polymerase, the envelope gene (E) and the nucleocapsid gene (N), as described in international guidelines (Corman *et al.* [14]). It was used as our routine test during the day.

The Xpert Xpress SARS-CoV-2 assay (Cepheid) and the GenMark ePlex SARS-CoV-2 assay are sample-to-answer molecular diagnostic platforms. The former is based on the detection of 2 gene targets, N2 and E, according to an algorithm in parallel with the sample process control, and the latter is based on the detection of the N gene of SARS-CoV-2. They were used during the night, on weekends and for urgent results.

All data (clinical, laboratory, from both mothers and newborns) were prospectively collected from medical records.

All statistical analyses were performed using R software (Version 3.5.1, R Core Team [2018] https://www.R-project.org/). Continuous variables are expressed as medians with the interquartile range (IQR). Categorical variables are expressed as numbers with percentages. A two-tailed Mann-Whitney U test was used for statistical analysis of continuous variables. Fisher's exact test was used for statistical analysis of categorical variables. Statistical significance was considered with a $p < 0.05$.

This study was approved by the institutional review board of the French College of Obstetricians and Gynecologists (2020-OBST-0408) and was performed in accordance with the Declaration of Helsinki. All patients gave written consent. All data were de-identified to ensure patient privacy and confidentiality.

## Results

During the study period, SARS-CoV-2 serology testing was offered to all 272 patients admitted to the delivery room (Fig 1). A total of 249 (91.5%) serology results were available (22 women did not give consent and one missing result). Seroprevalence was 8% (n = 20/249). The main characteristics of the women and of their pregnancies and newborns are shown in Table 1. All patients were asymptomatic on admission to the delivery room. There were no significant differences between the groups of SARS-CoV-2-IgG-positive and SARS-CoV-2-IgG-negative women. Among women admitted for delivery, 190 had a SARS-CoV-2 RT-PCR test during the study period (59 women declined testing and one missing result). Exhaustivity was therefore 69.9% (190/272). Only one (0.5%) test was positive and this patient remained asymptomatic during delivery and early post-partum. This positive test was performed on May 25, 2020, using the Xpert Xpress assay, with detection of the nucleocapsid (N2) gene only at a CT value of 41.4 within the valid range and endpoint above the minimum setting of this FDA-approved

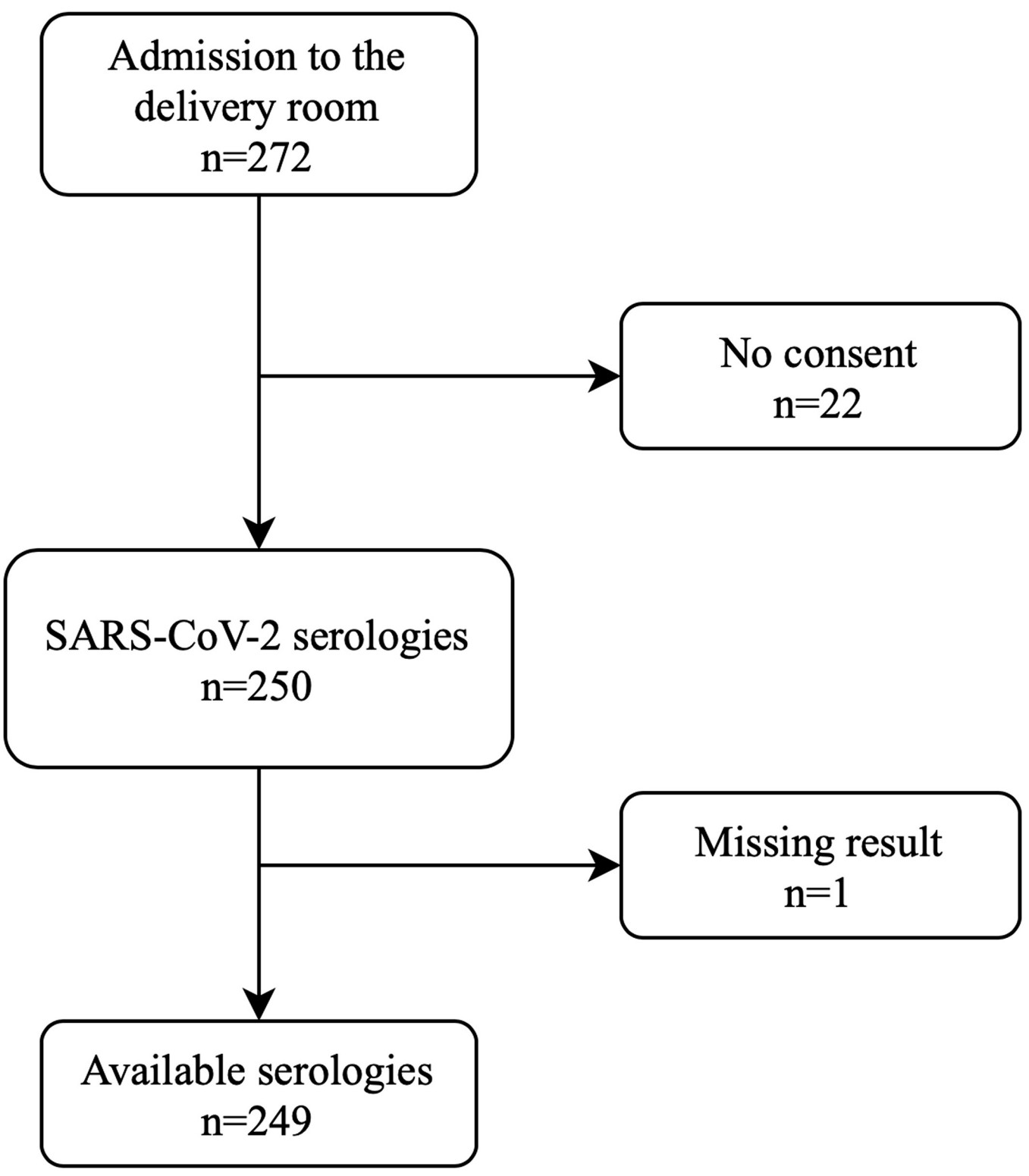

**Fig 1. Flow chart of the study–universal serology screening for SARS-CoV-2 among a cohort of women admitted for delivery.**

**Table 1. Maternal, obstetric and neonatal characteristics according to SARS-CoV-2 serological status for women admitted for delivery.**

|  | IgG-negative | IgG-positive | p |
|---|---|---|---|
|  | n = 229 | n = 20 |  |
| Age, years, median [IQR] | 33 [29–36] | 31 [30.5–37] | 0.88 |
| Nullipara, n (%) | 106 (46.3) | 12 (60) | 0.25 |
| BMI, kg/m$^2$, median [IQR] | 23 [21–27] | 25,5 [23–28.2] | 0.07 |
| History of preexisting: |  |  |  |
| - Diabetes mellitus, n (%) | 2 (0.9) | 0 (0) | 1 |
| - Chronic hypertension, n (%) | 6 (2.6) | 0 (0) | 1 |
| - Tobacco use, n (%) | 44 (19.2) | 2 (10.6) | 0.54 |
| - Asthma, n (%) | 13 (5.7) | 0 (0) | 0.6 |
| Gestational age at delivery, week median [IQR] | 39.6 [38.2–40.7] | 39.7 [38.2–40.3] | 0.50 |
| Delivery before 37 WG, n (%) | 26 (11.4) | 4 (21.1) | 0.26 |
| Birthweight, g median [IQR] | 3241 [2829–3585] | 3213 [2853–3471] | 0.62 |
| Infected neonates, n (%) | 0 (0) | 0 (0) |  |

assay. This indicates that the viral load of this sample is at the detection limit of the Xpert Xpress assay and may be related to an old infection. The serology was positive. The patient and her spouse reported symptoms such as headache, asthenia and anosmia as of March 10, 2020. Her newborn tested negative (negative SARS-CoV-2 RT-PCR performed on nasopharyngeal swabs at birth and at three days of life).

Analysis of the available questionnaires (n = 220/249; 88.3%) showed that 31.6% (6/19) of the SARS-CoV-2-IgG-positive women reported being asymptomatic throughout the first wave of the pandemic. SARS-CoV-2-IgG-positive women reported more symptoms compared to SARS-CoV-2-IgG-negative women (Table 2), such as fever (n = 3 (15.8%) versus n = 3 (1.5%); OR = 12.9 (95% CI 1.49–97.52); p = 0.009), myalgia (n = 7 (36.8%) versus n = 14 (7%); OR = 8.39 (95% CI 2.38–25.46); p<0.001) and anosmia (n = 6 (31.6%) versus n = 3 (1.5%); OR = 29.2 (95% CI 5.52–201); p<0.001). There was no difference regarding the symptoms of cough, dyspnea, diarrhea and rash. All women except one (in the negative group) reported they had strictly respected lockdown rules.

## Discussion

SARS-CoV-2 screening (serology and RT-PCR on nasopharyngeal secretions) of all women at delivery showed that: 1- the seroprevalence of SARS-CoV-2 infection was 8% and most of the pregnant women remained uninfected; 2- half of the SARS-CoV-2-IgG-positive pregnant women were completely asymptomatic during the study period; 3- the virus was no longer in active circulation up to three weeks after the end of lockdown.

In our maternity ward, implementation and generalized respect of lockdown rules and social distancing allowed for a drastic reduction in the number of confirmed cases. Pregnant women reported full lockdown compliance from March 17, 2020 until delivery. Despite the end of lockdown and the resumption of activity for members of the family circle, the rate of positive results by RT-PCR remained very low. In addition, the triad fever, myalgia and anosmia was linked with previous infection with the SARS-CoV-2 virus.

The seroprevalence of our cohort was similar to that observed in the general population [15, 16]. Crovetto and colleagues have recently reported a seroprevalence of 14% among 874 pregnant women [17]. However, this cohort included more than 40% tested in the 1$^{st}$ trimester of pregnancy. In addition, seroprevalence included positivity for IgG but also for IgM and IgA.

**Table 2. Reported signs and symptoms according to SARS-CoV-2 serological status for women admitted for delivery.**

|  | IgG-negative | IgG-positive | OR (95% CI) | p |
|---|---|---|---|---|
|  | n = 201 | n = 19 |  |  |
| Fever, n (%) | 3 (1.5) | 3 (15.8) | OR = 12.9 (1.49–97.52) | 0.009 |
| Cough, n (%) | 13 (6.5) | 2 (10.5) | OR = 1.69 (0.17–8.52) | 0.62 |
| Dyspnea, n (%) | 12 (6) | 1 (5.3) | OR = 0.87 (0.02–6.6) | 1 |
| Myalgia, n (%) | 14 (7) | 7 (36.8) | OR = 8.39 (2.19–25.46) | <0.001 |
| Anosmia, n (%) | 3 (1.5) | 6 (31.6) | OR = 29,2 (5.52–201) | <0.001 |
| Diarrhea, n (%) | 21 (10.4) | 1 (5.3) | OR = 0.47 (0.01–3.34) | 0,7 |
| Rash, n (%) | 5 (2.5) | 1 (5.3) | OR = 2.16 (0.04–20.9) | 0.42 |

It is interesting to note that 60% of the seropositive patients in their cohort were asymptomatic, which is higher than that observed in our study.

To date, it has not been assessed whether SARS-CoV-IgG, detected with currently available ELISA assays, are neutralizing antibodies and consequently confer protection. However, it is of major importance to have an estimation of the rate of pregnant women who have not already been infected, indeed the challenge of reducing severe forms in pregnant patients and decreasing obstetric iatrogenesis in mild forms will be directly determined by the potential number of patients infected in a second wave [10, 11]. Our study shows that there is a significant reservoir of future infections and that barrier measure recommendations, even when well respected by pregnant patients, were not enough to significantly reduce encounters with the virus in a population sensitive to infection prevention. This encourages us to suggest a reflection on specific measures to be implemented to protect this population, built on precise knowledge of the seroprevalence of COVID-19 in pregnant women [18, 19].

One could argue that the study period would need to be longer in order to fully estimate the effect of the end of the lockdown and the risk of a resumption of the pandemic (so-called second wave). The design of our study did not allow us to assess the existence of sequelae in our patient population, however, it has recently been shown that pregnancy in the 2nd and 3rd trimesters represents an independent risk factor for oxygen therapy, hospitalization in an intensive care unit and use of mechanical ventilation [20, 21]. It is important to note that pregnant women are subject to specific recommendations to limit the risk of infection and it appears that seroprevalence, although not high, is similar to that observed in the general population at the same time [19]. For these reasons, it is imperative that special attention be paid to this at-risk population in the event of a second wave of infections. Specific directed awareness should be given to these patients, particularly about their participation in family gatherings in closed environments, which seems to be an important source of viral transmission. We believe that our study provides an important snapshot at a given point in time of the exposure to COVID-19 in an at-risk population in a region highly exposed to the virus during spring.

## Conclusion

Three weeks after the end of French lockdown, SARS-CoV-2 infections were scarce in our region. A very high proportion of SARS-CoV-2-IgG-negative pregnant women, which is comparable to that of the general population, must be taken into consideration in the event of a resurgence of the pandemic. The traces of a past active circulation of the virus in this fragile population during the spring wave should encourage public health authorities to take specific measures for this independent at-risk group, in order to reduce viral circulation in pregnant patients.

## Supporting information

**S1 Data.**
(XLSX)

## Acknowledgments

We would like to thank David Marsh for language editing.

## Author Contributions

**Conceptualization:** Alexandra Benachi.

**Data curation:** Hoda Zakaria.

**Formal analysis:** Jérémie Mattern.

**Investigation:** Jérémie Mattern, Christelle Vauloup-Fellous, Hoda Zakaria, Alexandra Benachi, Julie Carrara, Alexandra Letourneau, Nadège Bourgeois-Nicolaos, Daniele De Luca, Florence Doucet-Populaire.

**Methodology:** Alexandra Benachi, Alexandre J. Vivanti.

**Supervision:** Alexandra Benachi, Alexandre J. Vivanti.

**Validation:** Christelle Vauloup-Fellous, Nadège Bourgeois-Nicolaos, Florence Doucet-Populaire.

**Writing – original draft:** Alexandre J. Vivanti.

**Writing – review & editing:** Jérémie Mattern, Christelle Vauloup-Fellous, Hoda Zakaria, Alexandra Benachi, Julie Carrara, Alexandra Letourneau, Nadège Bourgeois-Nicolaos, Daniele De Luca, Florence Doucet-Populaire.

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
