## [Decision Letter · Decision Letter 0]

7 Aug 2020

PONE-D-20-20995

Post lockdown COVID-19 seroprevalence and circulation at the time of delivery, France

PLOS ONE

Dear Dr. Vivanti,

Thank you for submitting your manuscript to PLOS ONE. After careful consideration, we feel that it has merit but does not fully meet PLOS ONE’s publication criteria as it currently stands. Therefore, we invite you to submit a revised version of the manuscript that addresses the points raised during the review process.

Four experts in the field handled your manuscript. We are thankful for their time and efforts. Although some interest was found in your study, several major comments arose that overshadowed this enthusiasm. These comments include that: several statements need citations; there are concerns about specifics of diagnosis and testing for COVID-19; numerous questions about the study design, including how the women were experimentally grouped and sensitivity of assays; the data presentation needs work, including the inclusion of data for the local evaluations; the discussion needs to be more developed; and the conclusions should better reflect the findings. ALL of the reviewers' comments must be addressed in your revised manuscript.

We look forward to receiving your revised manuscript.

Kind regards,

Frank T. Spradley

Academic Editor

PLOS ONE

a) If there are ethical or legal restrictions on sharing a de-identified data set, please explain them in detail (e.g., data contain potentially identifying or sensitive patient information) and who has imposed them (e.g., an ethics committee). Please also provide contact information for a data access committee, ethics committee, or other institutional body to which data requests may be sent

Reviewers' comments:

Reviewer's Responses to Questions

**Comments to the Author**

1. Is the manuscript technically sound, and do the data support the conclusions?

Reviewer #1: Partly

Reviewer #2: Yes

Reviewer #3: Partly

Reviewer #4: Yes

2. Has the statistical analysis been performed appropriately and rigorously? 

Reviewer #1: Yes

Reviewer #2: Yes

Reviewer #3: Yes

Reviewer #4: Yes

3. Have the authors made all data underlying the findings in their manuscript fully available?

Reviewer #1: Yes

Reviewer #2: Yes

Reviewer #3: No

Reviewer #4: No

4. Is the manuscript presented in an intelligible fashion and written in standard English?

Reviewer #1: Yes

Reviewer #2: Yes

Reviewer #3: Yes

Reviewer #4: Yes

5. Review Comments to the Author

Reviewer #1: The background and methods of this study are very good. It is impressive that you have managed to collect this amount of data for such a specific patient demographic. The data analysis has also been performed well.

I feel however that the discussion needs to have more detail. Furthermore, you have only written your conclusion in the abstract; you also need a conclusion section in the main body of the manuscript.

The conclusion you have reached is that SARS-Cov-2 negative women need to be taken into consideration in case of a second wave. However, I feel that you need more justification of this stance. Your results state that 47.4% of the positive women did not have any symptoms and those who did have symptoms appear relatively mild. Furthermore, your results also state that the majority of women were uninfected.

Did you find any serious sequelae of SARS-CoV-2 infection in your study as a whole? If yes, you need to include this. If not, then how have you come to the conclusion that these patients require more consideration than the general population?

In summary, you need to do further work in your discussion, because your actual results do not really justify your conclusion. You need to elaborate further on the potential consequences of SARS-CoV-2 infection in pregnant women if this is the conclusion you have made, and this requires more research into maternal and neonatal morbidity and mortality due to SARS-CoV-2. You need to compare and contrast your results to other research on this topic, and write a discussion that properly debates whether pregnancy puts mothers and neonates at higher risk of death or not.

Reviewer #2: Review: “Post lockdown COVID-19 seroprevalence and circulation at the time of delivery,France”

General comments:

This is a research involving the first wave of the COVID-19 pandemic in Paris and the effect of lockdown on seroprevalence in your maternity ward, which has been significantly affected by the virus (more than 40 confirmed infections among pregnant women between March 12 and April 20). Pregnancy as a severity risk factor of severe acute respiratory syndrome coronavirus 2 77 (SARS-CoV-2) infection is still debated.This study prospectively to evaluate the impact of the first wave of the COVID-19 pandemic at the Antoine Béclère Hospital maternity ward (Paris area, France) from May 4 (one week before the end of lockdown) to May 31, 2020 86 (three weeks after the end of lockdown). I recommend a publication of this paper with major revisions.

Abstract:

1.Study design:“All patients admitted to the delivery room during this period were offered a SARS-CoV-2 serology test as well concomitant SARS-CoV-2 RT-PCR on a nasopharyngeal sample”.As we all known, Throat swab nucleic acid positive rate is low,how many times do your study deign If the first nucleic acid test is negative?

2.Result: “A history of symptoms during the epidemic, such as fever, 54 myalgia and anosmia, was suggestive of previous infection.” What is the distribution of various clinical symptoms.

Key words: Please add on.

Introduction:

1.The introduction is too simple to state the current studies of COVID-19.

2.“more than 40 confirmed infections among pregnant women between March 12 and April 20”，Do you have some references?

Materials and Methods:

1.“All patients admitted to the delivery room during this period were offered a SARS-CoV-2 serology test as well as SARS-CoV-2 RT-PCR on a nasopharyngeal sample”. Did the serological testing include IgM and why it is’t mentioned in the full text?

2.“All data (clinical, laboratory, from both mothers and newborns) were prospectively collected from medical records.” This is a prospective study. Have you list which laboratory indicators need to be tested?

Results:

1. “Only one (0.5%) test was positive and this 143 patient remained asymptomatic during delivery and early post-partum”, Does this patient have a CT scan and how about the Image performance?

Reviewer #3: “Post lockdown COVID-19 seroprevalence and circulation at the time of 1

delivery, France” reported that the positive rates of SARS-COV2 IgG and virus RNA of throat samples in group of patients in maternity ward around the end of LOCKdown timepoints. Although the data is convincing, it does not look conclusive since no data was shown the positive rates before the end of lockdown(ie. 14 days before lockdown; lockdown periods, postlockdown). Secondly, except that the only diagnosed patients were both RNAand IgG positive and had symptoms and epidemiological evidence, no current infections of SARS-COV-2 were completely tested for all other patients since no IgM test were performed(sensitivity of RNA detection is usually low). So , the conclusion that the virus was no longer circulating up to three weeks after the end of the lockdown may not be acurate. Thirdly. “ half of the SARS-CoV-2-IgG-positive pregnant women were completely asymptomatic “may not be acurate since a longer time (more than 5 months )of history of symptoms may be required. Therefore I do not think the manuscript merit to publish in Plos one Journal unless these issues are resolved. However, the conclusion that “a high proportion of SARS-CoV-2-IgG-negative pregnant women must be taken into into consideration in the event of a resurgence of the pandemic...” is helpful.

Reviewer #4: The authors have investigated an important issue for the field of obstetrics during the current pandemic. The reasoning behind and the research set up is sound. The results have additional value and provide insights into the clinical presentation and pathogenesis of COVID19 during pregnancy.

Comments:

Line 71-72: reference is missing

Line 73: reference is missing

Line 77-78 Please do explain the third trimester is a consensual factor for SARS-CoV-2 infection, including reference.

Line 79: the effect of the lockdown: how is the effect exactly investigated or evaluated?

Line 99: why not show the data of the local evaluation. This will added to the impact of the results. Is there a control group, on what is the threshold of < 0.8 based?

Lin 106: what is Cepheid, where is it based?

Line 107: why use different assay? What is the difference in sensitivity between assay or it there none?

Line 145: perhaps elaborate on the CT value, based on the used assay.

Line 161: how come 3 women report anosmia (in general)? Does this occur often in Paris?

Line 164: which group did the woman who did not respect the lockdown belong to?

Line 173: were they also asymptomatic during possible infection? With positive seroprevalence, one can assume she was post-infection. Meaning: did they report symptoms at any time? Do pregnant women have asymptomatic infection?

Discussion. There should be more details about the IgG-positive women and possible presentation, effect, risks and implications for pregnancy and hypotheses behind. What are the recommendation at a second wave based on the results.

6. PLOS authors have the option to publish the peer review history of their article (what does this mean?). If published, this will include your full peer review and any attached files.

Reviewer #1: **Yes: **Shuliang Oliver Cheng

Reviewer #2: No

Reviewer #3: No

Reviewer #4: No

---

## [Author Response · Author response to Decision Letter 0]

10 Sep 2020

Clamart, September 7, 2020

Dear Editor,

Thank you for the opportunity to revise our paper. On behalf of all authors, I’m sending you a revised version of our manuscript entitled “Post lockdown COVID-19 seroprevalence and circulation at the time of delivery, France”.

We have carefully considered the suggestions and comments of the reviewers to improve the quality of our work. We are willing to provide other changes or clarification if you so wish.

We hope that this revised version of the manuscript will still be considered for publication in Plos One.

Thank you again for your work,

Yours sincerely,

Alexandre VIVANTI (MD, PhD)

Reviewer #1

The background and methods of this study are very good. It is impressive that you have managed to collect this amount of data for such a specific patient demographic. The data analysis has also been performed well.

I feel however that the discussion needs to have more detail. Furthermore, you have only written your conclusion in the abstract; you also need a conclusion section in the main body of the manuscript.

Thank you for your comments. We have added a conclusion to the main text of the manuscript.

The conclusion you have reached is that SARS-Cov-2 negative women need to be taken into consideration in case of a second wave. However, I feel that you need more justification of this stance. Your results state that 47.4% of the positive women did not have any symptoms and those who did have symptoms appear relatively mild. Furthermore, your results also state that the majority of women were uninfected.

Did you find any serious sequelae of SARS-CoV-2 infection in your study as a whole? If yes, you need to include this. If not, then how have you come to the conclusion that these patients require more consideration than the general population?

The design of our study did not allow us to assess the existence of sequelae in our patient population. However, it has recently been shown that pregnancy in the 2nd and 3rd trimesters represents an independent risk factor for oxygen therapy, hospitalization in an intensive care unit and use of mechanical ventilation (1,2). It is important to note that pregnant women are subject to specific recommendations to limit the risk of infection and it appears that seroprevalence, although not high, is similar to that observed in the general population at the same time. For these reasons, it is imperative that special attention be paid to this at-risk population in the event of a second wave of infections. We have provided further details in the manuscript.

1) Ellington S, Strid P, Tong VT, Woodworth K, Galang RR, Zambrano LD, et al. Characteristics of Women of Reproductive Age with Laboratory-Confirmed SARS-CoV-2 Infection by Pregnancy Status — United States, January 22–June 7, 2020. MMWR Morb Mortal Wkly Rep. 2020;69: 769–775. doi:10.15585/mmwr.mm6925a1

2) Badr DA, Mattern J, Carlin A, Cordier A-G, Maillart E, Hachem LE, et al. Are clinical outcomes worse for pregnant women ≥ 20 weeks’ gestation infected with COVID-19? A multicenter case-control study with propensity score matching. American Journal of Obstetrics & Gynecology. 2020;0. doi:10.1016/j.ajog.2020.07.045

In summary, you need to do further work in your discussion, because your actual results do not really justify your conclusion. You need to elaborate further on the potential consequences of SARS-CoV-2 infection in pregnant women if this is the conclusion you have made, and this requires more research into maternal and neonatal morbidity and mortality due to SARS-CoV-2. You need to compare and contrast your results to other research on this topic, and write a discussion that properly debates whether pregnancy puts mothers and neonates at higher risk of death or not.

Consistent with your suggestions, we have enhanced the discussion in relation to recent data in the literature.

Reviewer #2

Review: “Post lockdown COVID-19 seroprevalence and circulation at the time of delivery,France”

General comments:

This is a research involving the first wave of the COVID-19 pandemic in Paris and the effect of lockdown on seroprevalence in your maternity ward, which has been significantly affected by the virus (more than 40 confirmed infections among pregnant women between March 12 and April 20). Pregnancy as a severity risk factor of severe acute respiratory syndrome coronavirus 2 77 (SARS-CoV-2) infection is still debated. This study prospectively to evaluate the impact of the first wave of the COVID-19 pandemic at the Antoine Béclère Hospital maternity ward (Paris area, France) from May 4 (one week before the end of lockdown) to May 31, 2020 86 (three weeks after the end of lockdown). I recommend a publication of this paper with major revisions.

Abstract:

1.Study design:“All patients admitted to the delivery room during this period were offered a SARS-CoV-2 serology test as well concomitant SARS-CoV-2 RT-PCR on a nasopharyngeal sample”. As we all known, Throat swab nucleic acid positive rate is low,how many times do your study deign If the first nucleic acid test is negative?

Indeed, the sensitivity of the PCR is considered average (60-80%). As this is a routine screening in a population of asymptomatic women, we performed only one nasopharyngeal swab. In our opinion, multiple tests can be considered when there is a strong clinical presumption.

We have added details to the abstract.

2.Result: “A history of symptoms during the epidemic, such as fever, 54 myalgia and anosmia, was suggestive of previous infection.” What is the distribution of various clinical symptoms.

A history of symptoms during the epidemic, such as fever (15.8%), myalgia (36.8%) and anosmia (31.6%), was suggestive of previous infection.

Details added to the abstract

Key words: Please add on.

Keywords added: Pregnancy, SARS-CoV-2, Covid-19, Seroprevalence, RT-PCR

Introduction:

1.The introduction is too simple to state the current studies of COVID-19.

We have provided more information about the issues that justify a seroprevalence study in pregnant women.

2.“more than 40 confirmed infections among pregnant women between March 12 and April 20”，Do you have some references?

Pregnant patients in our maternity ward who were infected with SARS-CoV-2 were the subject of a publication.

We have provided the reference in the manuscript.

Vivanti AJ, Mattern J, Vauloup-Fellous C, Jani J, Rigonnot L, El Hachem L, et al. Retrospective description of 100 pregnant women with SARS-CoV-2 infection, France. Emerg Infect Dis. 2020.

Materials and Methods:

1.“All patients admitted to the delivery room during this period were offered a SARS-CoV-2 serology test as well as SARS-CoV-2 RT-PCR on a nasopharyngeal sample”. Did the serological testing include IgM and why it is’t mentioned in the full text?

The serological test did not not include IgM. Mention added.

2.“All data (clinical, laboratory, from both mothers and newborns) were prospectively collected from medical records.” This is a prospective study. Have you list which laboratory indicators need to be tested?

Only serological data and RT-PCR results were collected.

Results:

1. “Only one (0.5%) test was positive and this 143 patient remained asymptomatic during delivery and early post-partum”, Does this patient have a CT scan and how about the Image performance?

The patient did not have a CT scan. In our center, patients with a positive RT-PCR test with no sign of severity do not undergo a CT scan.

The design of our study does not allow us to evaluate the performance of the scanner in pregnant women infected with COVID-19.

Reviewer #3

“Post lockdown COVID-19 seroprevalence and circulation at the time of 1

delivery, France” reported that the positive rates of SARS-COV2 IgG and virus RNA of throat samples in group of patients in maternity ward around the end of LOCKdown timepoints. 

Although the data is convincing, it does not look conclusive since no data was shown the positive rates before the end of lockdownc(ie. 14 days before lockdown; lockdown periods, postlockdown).

We didn’t test all the patients during the lockdown (only symptomatic women), so we didn’t have the previous positive rate, but we assumed that this one was much higher.

 Secondly, except that the only diagnosed patients were both RNA and IgG positive and had symptoms and epidemiological evidence, no current infections of SARS-COV-2 were completely tested for all other patients since no IgM test were performed (sensitivity of RNA detection is usually low). So, the conclusion that the virus was no longer circulating up to three weeks after the end of the lockdown may not be acurate.

You are right, RT-PCR sensitivity must be taken into account. However, we used the same RT-PCR test that resulted in the diagnosis of over 40 patients with symptoms related to the 15/03 -18/04 period. Even taking into account a sensitivity of 60% of the RT-PCR test (low range), one could imagine that the detection of IgM could have allowed diagnosis for one more patient.

We have, however, tempered the conclusions regarding the non-circulation of the virus.

Thirdly. “ half of the SARS-CoV-2-IgG-positive pregnant women were completely asymptomatic “may not be acurate since a longer time (more than 5 months) of history of symptoms may be required. 

We do not fully agree with this assertion, since to date no robust published data have defined the monitoring period that enables us to assert the absence of symptoms/sequelae.

Consistent with your suggestion, we mention that the absence of symptoms was related to the duration of the study.

Therefore I do not think the manuscript merit to publish in Plos one Journal unless these issues are resolved. However, the conclusion that “a high proportion of SARS-CoV-2-IgG-negative pregnant women must be taken into into consideration in the event of a resurgence of the pandemic...” is helpful.

Reviewer #4

The authors have investigated an important issue for the field of obstetrics during the current pandemic. The reasoning behind and the research set up is sound. The results have additional value and provide insights into the clinical presentation and pathogenesis of COVID19 during pregnancy.

Comments:

Line 71-72: reference is missing 

Added

Line 73: reference is missing

Added

Line 77-78 Please do explain the third trimester is a consensual factor for SARS-CoV-2 infection, including reference.

It has recently been shown that pregnancy in the 2nd and 3rd trimesters represents an independent risk factor for oxygen therapy, hospitalization in an intensive care unit and use of mechanical ventilation.

These data are from case-control studies using a propensity score, which allows a robust response regarding the involvement of pregnancy as an independent risk factor.

1) Ellington S, Strid P, Tong VT, Woodworth K, Galang RR, Zambrano LD, et al. Characteristics of Women of Reproductive Age with Laboratory-Confirmed SARS-CoV-2 Infection by Pregnancy Status — United States, January 22–June 7, 2020. MMWR Morb Mortal Wkly Rep. 2020;69: 769–775. doi:10.15585/mmwr.mm6925a1

2) Badr DA, Mattern J, Carlin A, Cordier A-G, Maillart E, Hachem LE, et al. Are clinical outcomes worse for pregnant women ≥ 20 weeks’ gestation infected with COVID-19? A multicenter case-control study with propensity score matching. American Journal of Obstetrics & Gynecology. 2020;0. doi:10.1016/j.ajog.2020.07.045

Line 79: the effect of the lockdown: how is the effect exactly investigated or evaluated?

Lockdown has been shown to result in a very significant decrease in the incidence of COVID-19 infections in pregnant women in France.

Kayem G, Alessandrini V, Azria E, Blanc J, Bohec C, Bornes M, et al. A snapshot of the Covid-19 pandemic among pregnant women in France. Journal of Gynecology Obstetrics and Human Reproduction. 2020; 101826. doi:10.1016/j.jogoh.2020.101826

We aimed to evaluate the effects of the lifting of lockdown on a possible increase of the incidence of COVID-19 infection rates.

We provide extra information throughout the manuscript.

Line 99: why not show the data of the local evaluation. This will added to the impact of the results. Is there a control group, on what is the threshold of < 0.8 based?

The data are from a large cohort of patients. A control group is required to set a threshold.

These data will be submitted for publication. A manuscript is in preparation

Lin 106: what is Cepheid, where is it based?

Cepheid is an American molecular diagnostics company based in California, USA.

We have detailed this in the manuscript.

Line 107: why use different assay? What is the difference in sensitivity between assay or it there none?

We adapted our algorithm of the assay choice to our workflow, to our reagent supply possibilities and to the urgency of the results for clinical management of some parturients.

We now mention this in the manuscript 

The GeneFinder assay was used as our routine test during the day and this technique requires personnel qualified in molecular biology and a relatively high turnaround time (4 hours).

We specify this in the manuscript: It was used as our routine test during the day.

The Xpert Xpress SARS-CoV-2 assay (Cepheid) and the GenMark ePlex SARS-CoV-2 assay (ELItech) are two platforms that integrate specimen processing, nucleic acid extraction and reverse transcription PCR. Samples can be tested as soon as they are received, and results are generated in about 45 min. They were used during the night, on weekends and for urgent results. 

We now mention this in the manuscript 

All assays used have the same analytical performances.

Line 145: perhaps elaborate on the CT value, based on the used assay.

Only one PCR was positive and only nucleocapsid (N2) gene was detected at a CT value of 41.4 within the valid range and an endpoint above the minimum setting of this FDA-approved assay. This indicates that the viral load of the sample is at the limit of detection of the Xpert Xpress assay. 

We now mention this in the manuscript 

Loeffelholz MJ, Alland D, Butler-Wu SM, Pandey U, Perno CF, Nava A, Carroll KC, Mostafa H, Davies E, McEwan A, Rakeman JL, Fowler RC, Pawlotsky JM, Fourati S, Banik S, Banada PP, Swaminathan S, Chakravorty S, Kwiatkowski RW, Chu VC, Kop J, Gaur R, Sin MLY, Nguyen D, Singh S, Zhang N, Persing DH. Multicenter Evaluation of the Cepheid Xpert Xpress SARS-CoV-2 Test. J Clin Microbiol 2020 23;58(8):e00926-20. doi: 10.1128/JCM.00926-20.

Line 161: how come 3 women report anosmia (in general)? Does this occur often in Paris?

We have not particularly explored these specific patient responses. We do not have any data that could lead us to believe that anosmia often happens in Paris, but we can report that this symptom has been highly publicized in France. These reported symptoms may come from impressions reinforced by the strong media coverage of anosmia during the peak of the epidemic, making this benign symptom a potential marker of an infection encountered just before lockdown.

It could also be the result of false-negative IgG test 

Line 164: which group did the woman who did not respect the lockdown belong to?

She belonged to the negative group (now specified).

Line 173: were they also asymptomatic during possible infection? With positive seroprevalence, one can assume she was post-infection. Meaning: did they report symptoms at any time? Do pregnant women have asymptomatic infection?

Patients from this cohort have reported any symptoms occurring from January 1st.

We have no reason to think that pregnant patients have more asymptomatic infections than the general population. Rather, the tendency is to believe that pregnancy increases the severity of symptoms.

The test used to assess the presence of SARS-CoV-2 IgG is a commercial chemiluminescent immunoassay validated for clinical use by the FDA with high analytical and clinical performances.

Ref:

Bryan A, Pepper G, Wener MH, Fink SL, Morishima C, Chaudhary A, Jerome KR, Mathias PC, Greninger AL. Performance Characteristics of the Abbott Architect SARS-CoV-2 IgG Assay and Seroprevalence in Boise, Idaho. J Clin Microbiol. 2020 Jul 23;58(8):e00941-20.

EUA Authorized Serology Test Performance. https://www.fda.gov/medical-devices/coronavirus-disease-2019-covid-19-emergency-use-authorizations-medical-devices/eua-authorized-serology-test-performance) 

In just the past few months, multiple studies have characterized the kinetics of IgM and IgG, in the immune response to SARS-CoV-2 infection. IgM antibodies are readily produced as early as 1 day after infection, and the time of IgG seroconversion is expected to be in the range of 7 to 14 days after the onset of symptoms. However, Okba et al. reported that IgG and IgA kinetics depended on disease severity, and especially that patients with more severe disease developed an antibody response sooner and in higher concentrations. As our patients had mild symptoms, a late seroconversion is possible. Unfortunately, sera collected later were not available.

Ref:

Wang Y, Zhang L, Sang L. Kinetics of viral load and Antibody Responses in relation to COVID-19 severity. J Clin Invest. 2020; 138759. DOI: 10.1172/JCI138759.

Wölfel R, Corman VM, Guggemos W, et al. Virological assessment of hospitalized patients with COVID-2019. Nature 2020; 581(7809): 465-9. DOI: 10.1038/s41586-020-2196-x

Long QX, Liu BZ, Deng HJ, et al. Antibody responses to SARS-CoV-2 in patients with COVID19. Nature Med 2020; 26(6): 845-8. DOI: 10.1038/s41591-020-0897-1.

Okba NMA, Müller MA, Li W, et al. Severe Acute Respiratory Syndrome Coronavirus 2-Specific Antibody Responses in Coronavirus Disease Patients. Emerg Infect Dis. 2020; 26(7): 1478-88. DOI: 10.3201/eid2607.200841

Discussion. There should be more details about the IgG-positive women and possible presentation, effect, risks and implications for pregnancy and hypotheses behind. What are the recommendation at a second wave based on the results.

We have added extra explanations to the discussion section

---

## [Decision Letter · Decision Letter 1]

5 Oct 2020

Post lockdown COVID-19 seroprevalence and circulation at the time of delivery, France

PONE-D-20-20995R1

Dear Dr. Vivanti,

We’re pleased to inform you that your manuscript has been judged scientifically suitable for publication and will be formally accepted for publication once it meets all outstanding technical requirements.

Kind regards,

Frank T. Spradley

Academic Editor

PLOS ONE

Reviewers' comments:

Reviewer's Responses to Questions

**Comments to the Author**

1. If the authors have adequately addressed your comments raised in a previous round of review and you feel that this manuscript is now acceptable for publication, you may indicate that here to bypass the “Comments to the Author” section, enter your conflict of interest statement in the “Confidential to Editor” section, and submit your "Accept" recommendation.

Reviewer #1: All comments have been addressed

Reviewer #3: All comments have been addressed

2. Is the manuscript technically sound, and do the data support the conclusions?

Reviewer #1: Yes

Reviewer #3: Yes

3. Has the statistical analysis been performed appropriately and rigorously? 

Reviewer #1: Yes

Reviewer #3: Yes

4. Have the authors made all data underlying the findings in their manuscript fully available?

Reviewer #1: Yes

Reviewer #3: Yes

5. Is the manuscript presented in an intelligible fashion and written in standard English?

Reviewer #1: Yes

Reviewer #3: Yes

6. Review Comments to the Author

Reviewer #1: Thank you for adding to the discussion and highlighting recommendations that ought to be made to pregnant women to reduce transmission of SARS-CoV-2. Furthermore, thank you for highlighting the added risk due to pregnancy of severe sequelae of SARS-CoV-2 infection.

Reviewer #3: The author has answered all the questions and I recommend publication of this manuscript in Plos One

7. PLOS authors have the option to publish the peer review history of their article (what does this mean?). If published, this will include your full peer review and any attached files.

Reviewer #1: **Yes: **Shuliang Oliver Cheng

Reviewer #3: No

---

## [Editor Report · Acceptance letter]

7 Oct 2020

PONE-D-20-20995R1 

Post lockdown COVID-19 seroprevalence and circulation at the time of delivery, France 

Dear Dr. Vivanti:

I'm pleased to inform you that your manuscript has been deemed suitable for publication in PLOS ONE. Congratulations! Your manuscript is now with our production department. 

Kind regards, 

on behalf of

Dr. Frank T. Spradley 

Academic Editor

PLOS ONE